# Herpes simplex viral nucleoprotein creates a competitive transcriptional environment facilitating robust viral transcription and host shut off

Sarah E Dremel, Neal A DeLuca*

Department of Microbiology and Molecular Genetics, University of Pittsburgh School of Medicine, Pittsburgh, United States

**Abstract** Herpes simplex virus-1 (HSV-1) replicates within the nucleus coopting the host's RNA Polymerase II (Pol II) machinery for production of viral mRNAs culminating in host transcriptional shut off. The mechanism behind this rapid reprogramming of the host transcriptional environment is largely unknown. We identified ICP4 as responsible for preferential recruitment of the Pol II machinery to the viral genome. ICP4 is a viral nucleoprotein which binds double-stranded DNA. We determined ICP4 discriminately binds the viral genome due to the absence of cellular nucleosomes and high density of cognate binding sites. We posit that ICP4's ability to recruit not just Pol II, but also more limiting essential components, such as TBP and Mediator, create a competitive transcriptional environment. These distinguishing characteristics ultimately result in a rapid and efficient reprogramming of the host's transcriptional machinery, which does not occur in the absence of ICP4.

DOI: https://doi.org/10.7554/eLife.51109.001

## Introduction

Like most DNA viruses, the genome of Herpes simplex virus-1 (HSV-1) is transcribed by RNA Polymerase II (Pol II) (*Alwine et al., 1974*). Its approximately 85 genes (*McGeoch et al., 1988*; *McGeoch et al., 1986*; *McGeoch et al., 1985*) are transcribed in a temporally coordinated sequence, such that their protein products are expressed at the appropriate time in the life cycle of the virus (*Honess and Roizman, 1974a*; *Honess and Roizman, 1974b*; *Honess and Roizman, 1975*). Immediate early (IE) gene products enable the efficient expression of early (E) and late (L) genes. The protein products of E genes are mostly involved in DNA replication. DNA replication and IE proteins enable the efficient transcription of L genes, which encode the structural components of the virus. DNA replication licenses L promoters, enabling the binding of core Pol II transcription factors, thus activating the initiation of L transcription (*Dremel and DeLuca, 2019*). This entire transcriptional cascade is observed within 3 hour (h) post entry (*Dembowski and DeLuca, 2018*; *Dremel and DeLuca, 2019*), culminating in production of the first viral progeny between 4 and 6 h post-infection (hpi). To accomplish this robust and rapidly changing program of transcription, the viral genome must compete with the vastly larger cellular genome for numerous Pol II transcription factors, in addition to mediating the possible constraints of cellular histones.

A major component of this cascade is the IE protein Infected Cell Polypeptide 4 (ICP4) (*Courtney and Benyesh-Melnick, 1974*). ICP4 is essential for viral growth because it promotes efficient transcription of viral E and L genes (*Dixon and Schaffer, 1980*; *Preston, 1979*; *Watson and Clements, 1980*). Thus, in the absence of ICP4, E and L proteins are poorly expressed, IE proteins are overproduced, DNA replication does not occur, and there is no detectable viral yield

*For correspondence:
ndeluca@pitt.edu

Competing interests: The authors declare that no competing interests exist.

(*DeLuca et al., 1985*). ICP4 was first shown to bind to DNA cellulose made from salmon sperm DNA (*Powell and Purifoy, 1976*). *Faber and Wilcox (1986)* later showed ICP4 has sequence-specific DNA binding activity. ICP4 interacts with a number of cellular general transcription factors (GTFs), predominantly components of TFIID and the Mediator complex (*Carrozza and DeLuca, 1996*; *Lester and DeLuca, 2011*; *Wagner and DeLuca, 2013*), facilitating their recruitment to the viral genome through its DNA binding activity (*Dembowski and DeLuca, 2018*; *Lester and DeLuca, 2011*; *Sampath and Deluca, 2008*). ICP4 is synthesized early in infection, binds to the viral genome located at ND10 structures (*Everett et al., 2003*), and remains associated with the genome throughout all phases of infection (*Dembowski and DeLuca, 2018*). Therefore, ICP4 has the potential to influence events occurring on the viral genome from a time when genome number is at a minimum, and ICP4 expression is peaking, through a time when genome numbers are greatly elevated by replication.

Studies have also shown that epigenetic modulation of histones associated with the viral genome early in infection can affect productive viral infection (*Knipe and Cliffe, 2008*; *Liang et al., 2009*). However, we have shown that the abundance of histones is relatively low or absent, and that ICP4 is one of the most abundant proteins on viral genomes during productive infection (*Dembowski and DeLuca, 2015*). In this study, we set out to determine the relationship between ICP4 and histones binding to the viral and cellular genomes, and the consequences for viral and cellular transcription. We propose that ICP4 is a major component of viral nucleoprotein, which functions in place of traditional cellular chromatin, and allows for the robust recruitment of cellular transcription factors specifically to the viral genome.

## Results

### ICP4 binding is altered by viral genome replication

Given the central role of ICP4 in viral gene transcription at all stages of infection, we were interested in how ICP4 interacts with the virus genome as the number exponentially increases, as a consequence of replication. We infected human fibroblast (MRC5) cells with wild-type HSV-1 (KOS) for two, four, and six hpi and performed ChIP-Seq for ICP4. Each time point represents a different replication state: two hpi (prereplication), four hpi (3–4 genome duplications), six hpi (5–6 genome duplications) (*Figure 1A*). To quantitatively compare samples, we had to account for viral genome replication. Input samples provided the relative number of viral genomes present at each time point. We used this ratio to normalize immunoprecipitated (IP) sample for the amount of factor per genome. Early during infection (two hpi) ICP4 densely coated the viral genome (*Figure 1B–C*). Viral genome replication decreased the amount of ICP4 bound per genome (*Figure 1A–B*) resulting in a pattern containing sharper peaks. By six hpi ICP4 binding was retained exclusively on strong ICP4 binding motifs (*Figure 1C*). Some of the retained binding sites were those previously established as having an inhibitory effect on the gene promoter bound, including ORF P, ICP4, and LAT (*Figure 1D*). A closer analysis of ICP4 peaks demonstrated that the location and number of high confidence occupied sites did not alter significantly throughout infection (*Figure 1E*). Instead the amount of ICP4 bound between distinct peaks decreased as genome number increased (*Figure 1C–D*).

Although ICP4 exhibited a dense binding pattern at early times (two hpi), with relatively broad, overlapping peaks we were able to determine high confidence binding sites. The final sites were consistent between biological replicates. We analyzed 100 bp extensions from the summits of the peaks seen at all three times (*Figure 1E*) for motif discovery. DTSGKBDTBNHSG was the only motif discovered (*Figure 1F*), where D is A, G, T; S is C or G; K is G or T; B is C, G, T; H is A, C, T. This binding motif is similar to that previously discovered using in vitro techniques, RTCGTCNNYNYSG (*DiDonato et al., 1991*).

These results demonstrate that ICP4 binds to specific sites, but also coats the genome early in infection forming a type of nucleoprotein. Due to mass action, ICP4-nucleoprotein changes as infection proceeds, limiting binding to predominantly strong cognate binding sites as the number of genomes increase due to replication.

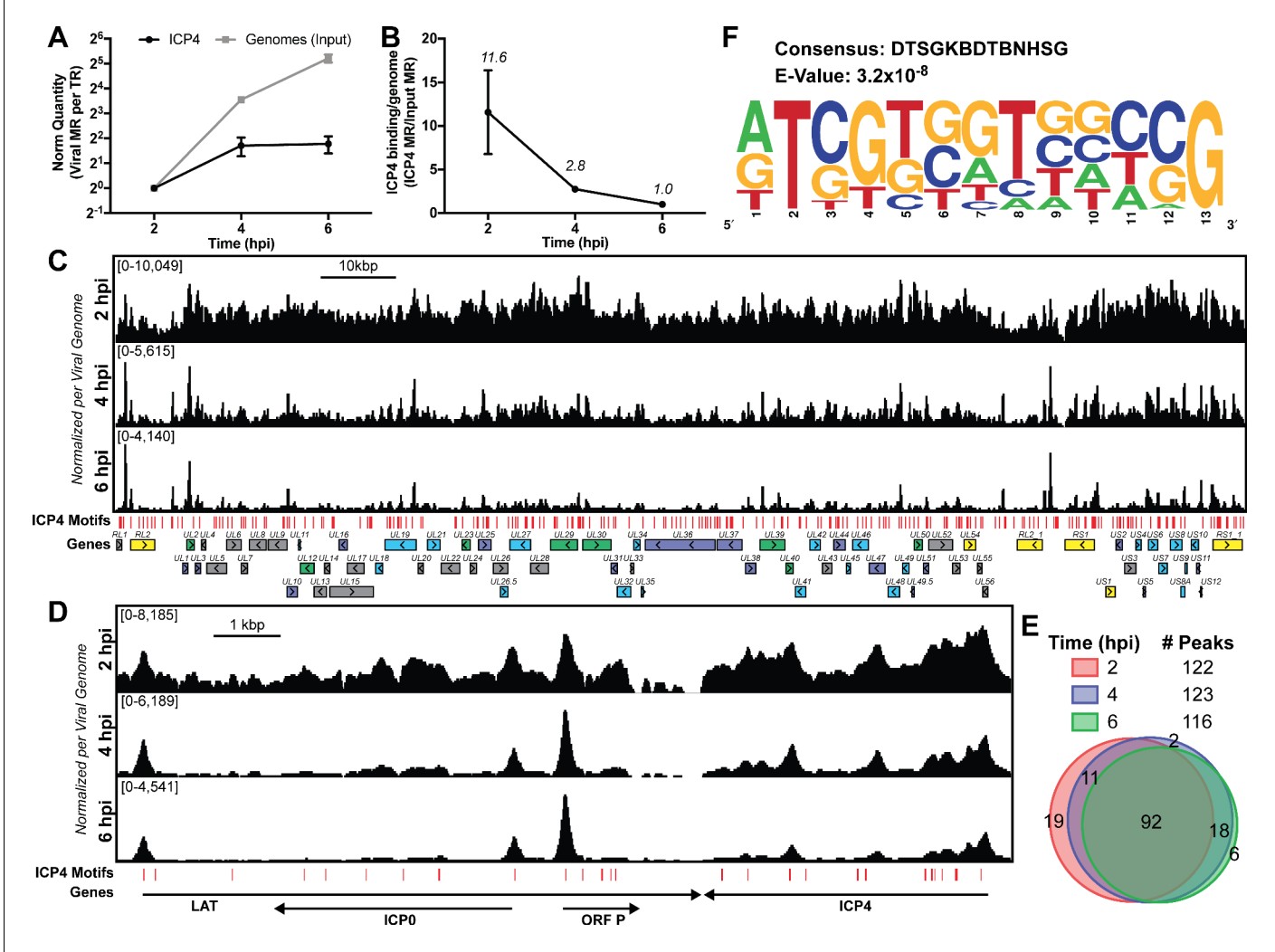

**Figure 1.** ICP4 binding at key points in the viral life cycle. MRC5 cells were infected with HSV-1 for 2, 4, or 6 h, and ChIP-Seq for ICP4 was performed. (A) Quantification of viral genomes, measured as Input sample viral mapped reads (MR), normalized for sequencing depth or Total Reads (TR). Samples were normalized to two hpi, which was set as 1. (B) Quantification of ICP4 binding per viral genome measured as viral MR from ICP4 immunoprecipitation (IP) per viral MR from Input. (C–D) All data was normalized for sequencing depth and viral genome number using input ChIP-Seq reads. Viral ORFs are indicated, color coded by gene class with IE as yellow, E as green, leaky late (L1) as blue, and true late (L2) as purple. Find Individual Motif Occurrences (FIMO) identified genome sequences matching the consensus motif in B are indicated in red. (E) Intersection of MACS2 identified ICP4 occupied regions. (F) ICP4 consensus binding motif.

DOI: https://doi.org/10.7554/eLife.51109.002

## ICP4 stabilizes GTF binding promoting cooperative preinitiation complex (PIC) assembly

We wanted to investigate how the formation of ICP4-nucleoprotein affects the transcription factor landscape across the viral genome. We compared the binding of ICP4, Pol II, TATA-binding protein (TBP), SP1, Med1, and Med23 in ICP4 null (n12) and wild-type (WT) HSV-1 infected human fibroblasts at 2.5 hpi by ChIP-Seq (*Figure 2—figure supplement 1*). In all Pol II IP's we used an antibody which preferentially binds to phospho S5 of the C-terminal domain repeat region. This post-translational modification is associated with Pol II found on mRNA promoters and splice sites.

In n12 infection, we observed a decrease in binding for all the factors to most viral promoters, with the exception of IE genes (*Figure 2A–B*), where there was an increase in binding. There were detectable, although highly reduced, peaks of TBP and SP1 binding to the UL23 promoter in the absence of ICP4. It has been previously shown that these sites are functional in the n12 background

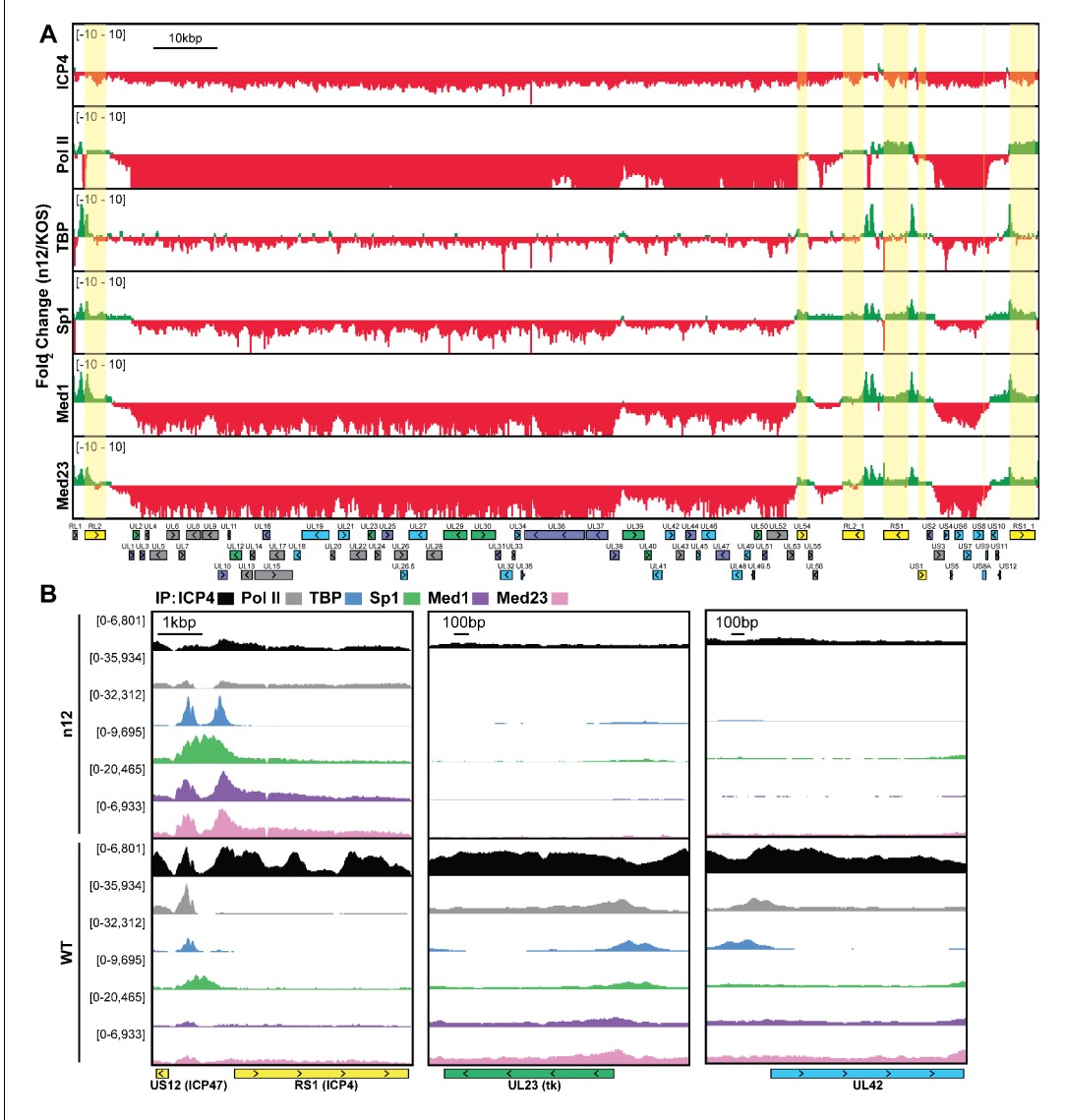

**Figure 2.** ICP4 recruitment of host Pol II machinery to viral promoters. MRC5 cells were infected with an ICP4 null mutant (n12) or HSV-1 (WT) for 2.5 h and ChIP-Seq for ICP4, Pol II, TBP, Sp1, Med1, and Med23 was performed. All data was normalized for sequencing depth and viral genome number using input ChIP-Seq reads. Viral ORFs are indicated, color coded by gene class with IE as yellow, E as green, leaky late (L1) as blue, and true late (L2) as purple. (**A**) Fold change of n12 over WT aligned to the viral genome. Loci with greater binding in n12 or WT are colored in green or red, respectively. (**B**) ChIP-Seq reads normalized per viral genome and aligned to canonical IE, E, and L1 genes.

DOI: https://doi.org/10.7554/eLife.51109.003

The following figure supplement is available for figure 2:

**Figure supplement 1.** MRC5 cells were infected with an ICP4 null mutant (n12) or HSV-1 (WT) for 2.5 h and ChIP-Seq for ICP4, Pol II, TBP, Sp1, Med1, and Med23 was performed.

DOI: https://doi.org/10.7554/eLife.51109.004

reflecting the basal binding activity of TBP and SP1 (*Imbalzano et al., 1991*) (*Figure 2B*). Similar to UL23 we observed TBP and SP1 bound to selective E promoters in the absence of ICP4, namely UL23, UL29, UL39, and UL50 (*Figure 2—figure supplement 1*).

There were a relatively small number of reads in the ICP4 ChIP of n12 (*Figure 2A*, *Figure 2—figure supplement 1*). As ICP4 is an essential viral protein, we prepared stocks of n12 in an ICP4 complementing cell line. ICP4 is packaged into the tegument of virions (*Yao and Courtney, 1989*), which results in packaging of wild type ICP4 into n12 virions (*Dembowski and DeLuca, 2018*). The

reads in the ICP4 ChIP of n12 are most likely due to ICP4 packaged in the virion. While the ultimate source of these reads is not clear at present, the amount of binding of ICP4 from the virion to DNA is not sufficient to promote transcription complexes on viral early and late genes.

Med1 and Med23 bound the viral genome with an almost identical pattern (*Figure 2—figure supplement 1*, *Figure 2*), indicating they are parts of the Mediator complex bound early during viral infection. In WT infected cells, the binding of Mediator concentrated near the starts sites of ICP4-induced viral genes. However, the Mediator complex also densely coated the viral genome, resembling the ICP4 binding pattern. This dense coating was completely absent in n12 infection, demonstrating this phenotype is not an artifact of the IP. We suspect this reflects the fact that ICP4 and Mediator interact.

In the absence of ICP4, the binding of Pol II was reduced the most compared to the other transcription factors (*Figure 2A*). This magnified difference is likely a result of the cooperative nature of Pol II recruitment requiring multiple protein-protein interactions. In summary ICP4 was required for robust recruitment of all GTF's tested, cooperatively recruiting Pol II to E and L promoters (*Figure 2B*). The difference between n12 and WT shows the extent by which ICP4 mediated recruitment and bolstered the frequency of PIC assembly. IE promoters retained robust GTF recruitment via an independent mechanism involving a complex consisting of Oct-1, HCF and VP16 binding to TAATGARAT promoter elements (*Preston et al., 1988*; *Stern and Herr, 1991*; *Stern et al., 1989*).

## Genome-bound ICP4 does not affect accessibility

Part of the mechanism of ICP4 action in the recruitment of GTFs to the genome may involve a role in the exclusion of repressive chromatin. To address this hypothesis, we investigated the relationship between presence of ICP4, the abundance of histones, and the accessibility of the genome. We used ChIP-Seq to compare the binding of ICP4, Pol II, and histone H3 in n12 and WT HSV-1 infected human fibroblasts at two hpi (*Figure 3*). We found in both WT and n12 infection that the number of H3 reads mapped to the viral genome was 100-fold less than ICP4, and the pattern was nearly identical to input reads (*Figure 3A*), with $R^2$ correlations of 0.0004 and 0.02 (*Figure 3—figure supplement 1*). These data demonstrated H3 binding to the viral genome was minimal and not reproducible. Furthermore, H3 binding was still minimal in the absence of ICP4 (n12). This was not due to technical issues as the number and quality of H3 reads mapped to the cellular genome for the same samples was approximately 10 million with $R^2$ correlations of $\geq 0.97$ (*Figure 3—figure supplement 2*). We saw a similar trend with H3K4me3, H3K27ac, H3K9me3, and H3K27me3 reads mapped to the viral genome (*Figure 3—figure supplements 1–3*).

Although H3 binding to the viral genome was similar in WT and n12 infection, we could not rule out the role of an alternative protein occluding the genome. To investigate genomic accessibility, we performed ATAC-Seq. Human fibroblasts were infected with WT and n12 HSV-1 at an MOI of 10 pfu/cell and collected prior to the onset of genome replication. Quantification of ChIP-Seq input reads allowed us to determine that the approximate number of genomes per cell in WT and n12 infection was 169 and 254, respectively (*Figure 3C*). This value is consistent with infecting at an MOI of 10 pfu/cell and an approximately particle to pfu ratio of 20–30. We normalized ATAC-Seq traces to adjust for sequencing depth and input genome number. We observed even tagmentation in both conditions (*Figure 3B*) absent the nucleosomal laddering visible on the cellular genome (*Figure 3D*). Quantification of ATAC-Seq reads determined that the viral genome in n12 and WT was 2.2 and 2-fold more accessible than the cellular genome (*Figure 3C*). As we harvested samples pre-replication, we expect that a significant portion of viral genomes are defective and will not undergo replication. Our ATAC-Seq data is thus an average of tagmentation for defective and active viral genomes. For this reason, we expect our accessibility calculation is an underestimate. We conclude that the viral genome was much more accessible than the cellular genome, and this increased accessibility was not ICP4-dependent. ICP4 binding and GTF recruitment, not viral genome accessibility, was responsible for robust GTF binding.

## ICP4 binds to cellular transcription start sites (TSS) early during infection

Immunofluorescence (IF) studies of HSV-1 infection depict colocalization of ICP4 with EdC-labeled viral genomes and exclusion from dense areas of cellular chromatin (*Dembowski and DeLuca,*

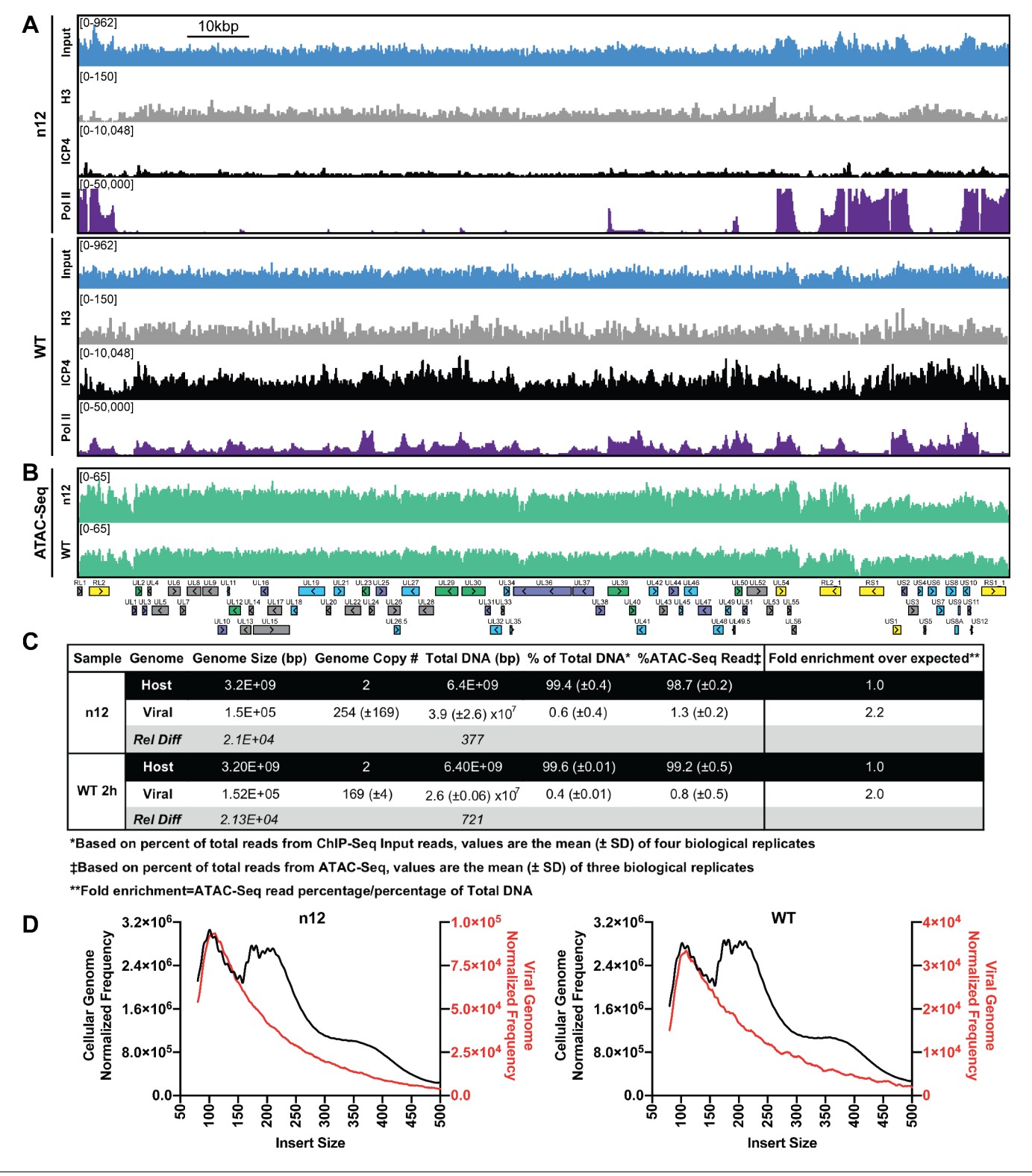

**Figure 3.** ICP4 dependence of viral genome accessibility and Histone H3 binding. MRC5 cells were infected with an ICP4 null mutant (n12) or HSV-1 (WT) and harvested prior to genome replication. (A) ChIP-Seq for ICP4, Pol II, H3 and B–D) ATAC-Seq was performed. All data was normalized for sequencing depth and viral genome number using input ChIP-Seq reads. (C) Quantitative analysis of ATAC-Seq data, measuring the relative tagmentation enrichment for the virus or host as compared to expected. (D) Histogram plot of ATAC-Seq fragment size for reads mapped to the viral (red) or cellular (black) genome. Mononucleosome protected fragments are approximately 180–250 bp.

*Figure 3 continued on next page*

*Figure 3 continued*

DOI: https://doi.org/10.7554/eLife.51109.005

The following figure supplements are available for figure 3:

**Figure supplement 1.** Analysis of ChIP-Seq data quality.

DOI: https://doi.org/10.7554/eLife.51109.006

**Figure supplement 2.** Analysis of ChIP-Seq data quality.

DOI: https://doi.org/10.7554/eLife.51109.007

**Figure supplement 3.** MRC5 cells were infected with an ICP4 null mutant (n12) or HSV-1 (WT) for 2 hr and ChIP-Seq for ICP4, Pol II, H3, H3K4me3, H3K27acetyl, H3K9me3, and H3K27me3 was performed.

DOI: https://doi.org/10.7554/eLife.51109.008

*2015*). This phenomenon is so well established that ICP4 is largely used in IF studies as a proxy for HSV-1 genomes. To ascertain if ICP4 also binds to the cellular genome, we aligned our ICP4 ChIP-Seq data from 2, 4, and six hpi to the cellular genome. ICP4 bound to the cellular genome in a manner quite distinctive from the pattern observed on the viral genome. ICP4 only bound in distinct peaks around cellular transcription start sites (TSS) of a subset of cellular genes (*Figure 4A–D*). These genes grouped ontologically to common housekeeping functions including pathways related to chromatin, transcription, and metabolism (*Figure 4F*). This binding reduced from 2 to 4 hpi, and

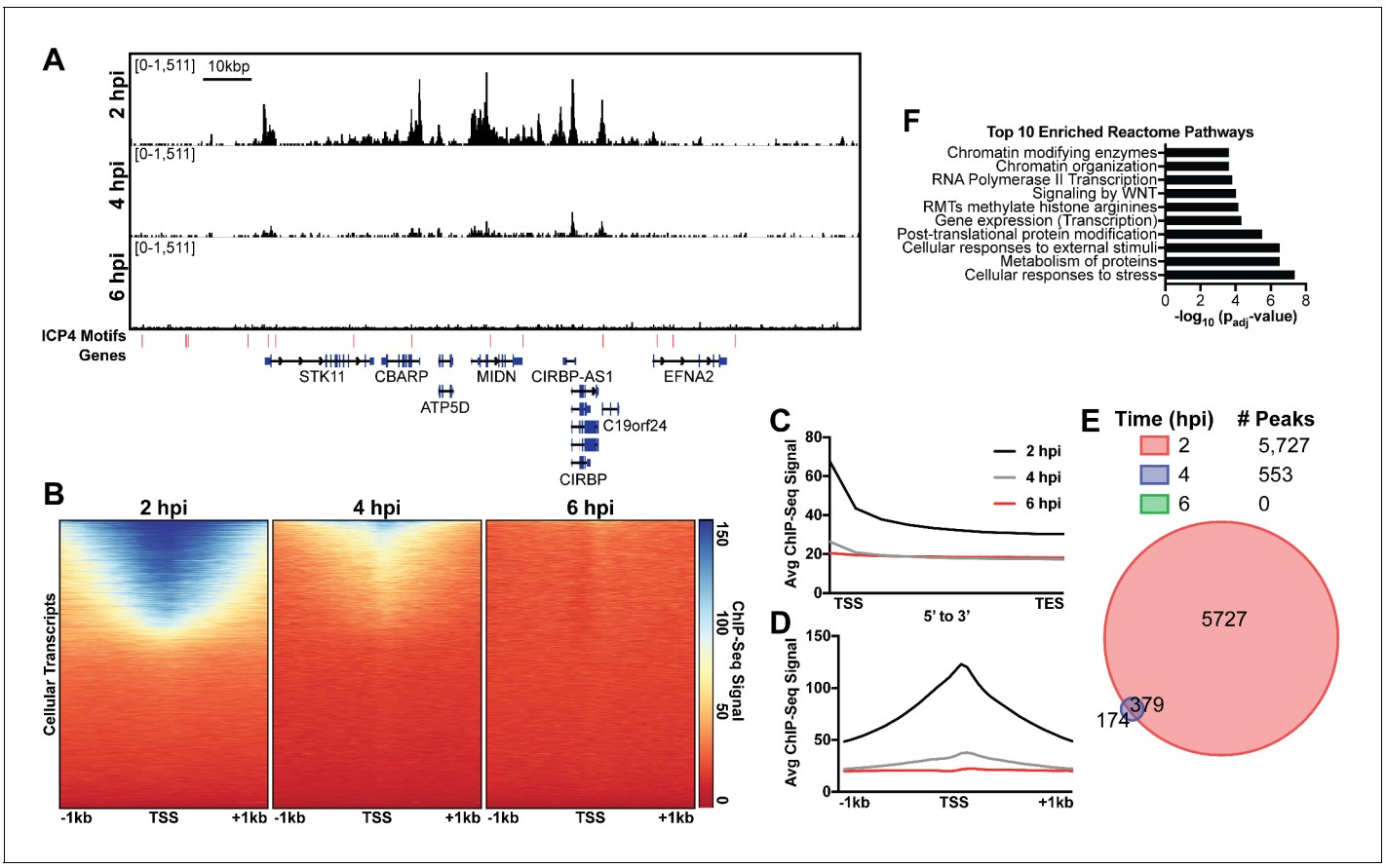

**Figure 4.** ICP4 binds cellular promoters during early infection. MRC5 cells were infected with HSV-1 for 2, 4, or 6 h, and ChIP-Seq for ICP4 was performed. Data was aligned to the human genome (hg38) and normalized for cellular genome sampling. (**A**) Representative region of the cellular genome. ICP4 binding motifs are indicated in red. (**B, D**) Sequencing data centered + /- 1 kilobase from the transcription start site (TSS) of cellular mRNAs. (**C**) Sequencing data scaled from mRNA TSS to the transcription end site (TES). (**E**) Intersection of MACS2 identified ICP4 occupied regions. (**F**) Top 10 enriched Reactome groups for genes (n = 2190) with ICP4 bound at the promoter at 2 hr.

DOI: https://doi.org/10.7554/eLife.51109.009

became negligible at six hpi (*Figure 4A–B*). At two hpi ICP4 bound to the cellular genome at 5727 sites (*Figure 4E*) or 0.002 peaks per kbp, whereas ICP4 bound to the viral genome at 122 sites (*Figure 1E*) or 0.8 peaks per kbp. Similar we found a much greater density of ICP4 binding motifs present in the viral genome (two motifs/kbp) than the cellular genome (0.02 motifs/kbp). We observed ICP4 binding peaks that did not localize at an ICP4 binding consensus (*Figure 4A*) suggesting that ICP4 may associate with the cellular genome by an alternative mechanism. We conclude that ICP4 bound to the cellular genome early during infection, when the relative concentration of ICP4 to viral genomes is still quite high. The amount of ICP4 on the cellular genome quickly dropped off as viral genome number increased and ICP4 preferentially bound to the viral genome.

## ICP4 binding is restricted to accessible regions of the cellular genome

Since ICP4 bound to a subset of cellular genes near mRNA start sites (*Figure 4*), We hypothesized that ICP4 only bound to accessible regions of the cellular genome. To test this hypothesis, we performed ChIP-Seq for ICP4, Pol II, Histone H3 (H3), euchromatic markers H3K4-trimethyl (H3K4me3) and H3K27-acetyl (H3K27ac), and heterochromatic marker H3K9-trimethyl (H3K9me3) and H3K27-trimethyl (H3K27me3) on MRC5 cells that were infected with HSV for 2 hr. Cellular TSS were stratified using k-means clustering as high and low ICP4 binding (*Figure 5A*). TSS with high ICP4 binding were also bound by Pol II and adjacent to euchromatic markers. TSS with low ICP4 binding were associated with only heterochromatic markers. Furthermore, genes clustered as high ICP4 binding had higher tagmentation frequency when assessed using ATAC-Seq (*Figure 5B*). The data was mapped for representative cellular genes in *Figure 5—figure supplement 1*. We quantified the relationship between ICP4 and cellular chromatin in *Figure 5C–D*. We found that the binding pattern of ICP4 was directly related (Spearman coefficient $\geq$0.5) to Pol II and cellular euchromatin, clustering as most similar to H3K27ac and H3K4me3 (*Figure 5C*). The heterochromatic markers, H3K9me3 and H3K27me3, clustered together, and were not correlated (Spearman coefficient ~0) to ICP4 or cellular chromatin. These results were corroborated by analysis of distinct peaks called using MACS (*Figure 5D*). Interestingly, ICP4 bound regions had little overlap with their cognate binding motifs (*Figure 5D*). A closer analysis of the actual genomic region where each factor bound, revealed that 82% of ICP4 bound regions were within 1 kb of a promoter (*Figure 5—figure supplement 2*). By comparison only 10% of ICP4 predicted binding motifs were within 1 kb of a promoter. Furthermore, the euchromatic regions of the cellular genome that were occupied by ICP4 in infected cells were also euchromatic in uninfected cells, indicating that ICP4 does not globally promote open chromatin in these regions of the genome (*Figure 5A–B*). These data support a model in which ICP4 is able to bind nonspecifically to accessible regions of the cellular genome, namely active promoters, early in infection when the relative concentration of ICP4 is high.

## ICP4 mediates depletion of pol II on cellular promoters

We observed depletion in Pol II binding to cellular promoters with infection (*Figure 5A*). This observation is consistent with prior studies, which assessed HSV-1 infection post-replication at three, four, or six hpi (*Abrisch et al., 2015*; *Birkenheuer et al., 2018*; *McSwiggen et al., 2019*). As we harvested samples prior to the onset of genome replication (two hpi) we hypothesized that ICP4, which is produced immediately upon viral infection was responsible. First, we determined the effect of ICP4 on cellular promoters before the onset of genome replication. We mock-infected or infected fibroblasts at 10 pfu/cell with WT or ICP4-null (n12) HSV-1 for 2 h. We observed depletion of Pol II occupancy on cellular mRNA promoters only in WT infection (*Figure 6A–B*). Thus we concluded that ICP4 was required for depletion of Pol II from host mRNA promoters, and this effect was independent of viral genome copy number.

We then assessed whether ICP4 was continuously required for cellular Pol II depletion, namely if ICP4 was still essential even after the onset of genome replication. We used a temperature sensitive ICP4 mutant (tsKos), in which growth at nonpermissive temperature (39.6°C) results in loss of ICP4 in the nucleus (*Dremel and DeLuca, 2019*). We infected fibroblasts with tsKos grown at permissive conditions (P), shifted up from permissive to nonpermissive conditions at four hpi (S), or nonpermissive conditions (N). In this system we can separate the role of ICP4 in Pol II depletion, from ICP4's requirement in E and L transcription and viral genome replication. Infected cells were harvested at four or six hpi and Pol II ChIP-Seq was performed. We used nonpermissive conditions as a surrogate

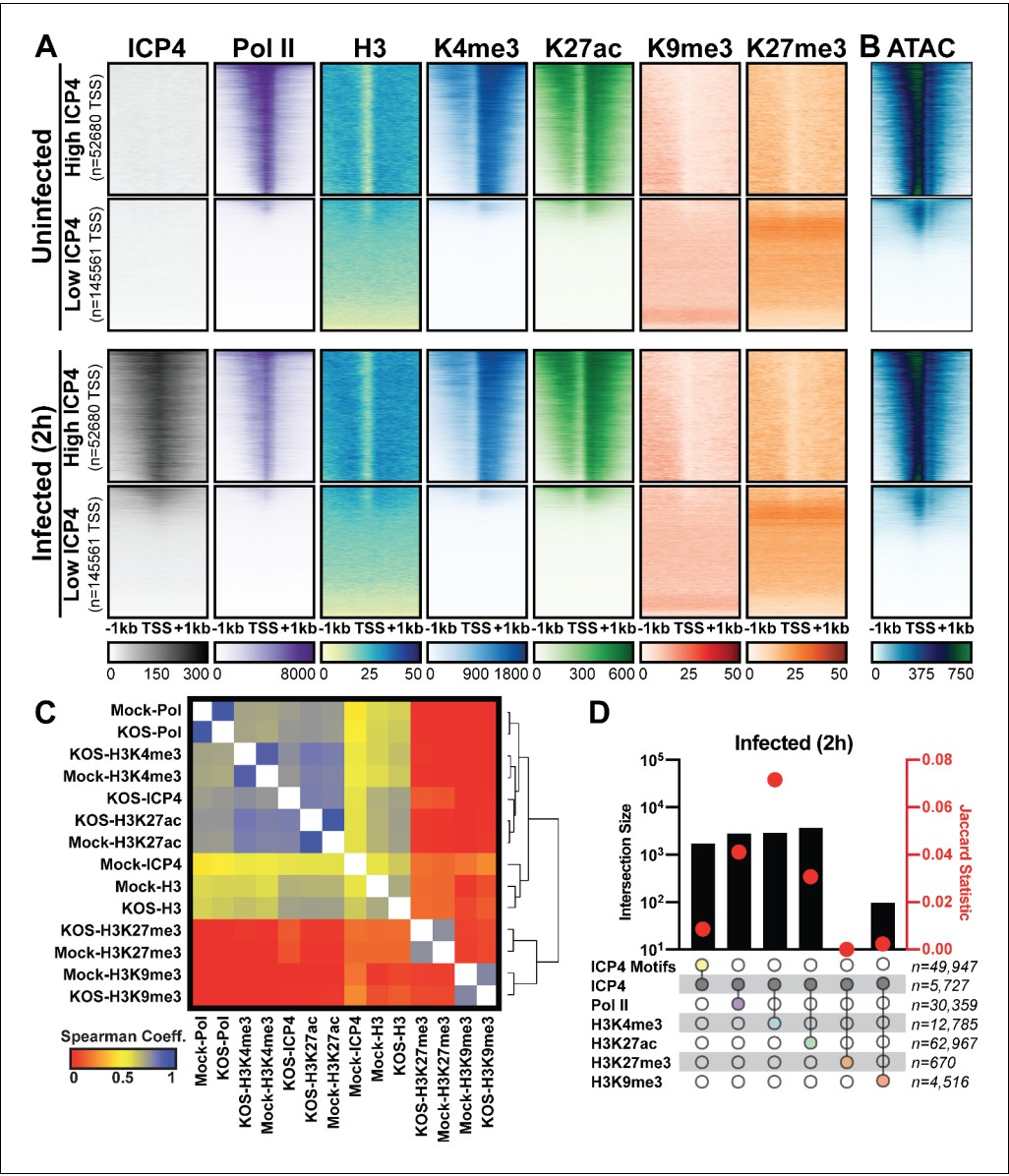

**Figure 5.** Association between cellular ICP4 binding and chromatin. MRC5 cells were uninfected or infected with HSV-1 for 2 h. All data was aligned to the human genome (hg38) and normalized for sequencing depth. (**A, C–D**) ChIP-Seq data for ICP4, Pol II, H3, H3K4me3, H3K27acetyl, H3K9me3, and H3K27me3. (**B**) ATAC-Seq data. (**A–B**) Sequencing data centered + /- 1 kilobase from the TSS of cellular mRNAs. Data was stratified for ICP4 binding using K-means clustering. (**C**) Spearman correlation analysis, limited to cellular transcripts. (**D**) Intersection of MACS2 peaks, analyzed as number of intersecting peaks or Jaccard statistic.

DOI: https://doi.org/10.7554/eLife.51109.010

The following figure supplements are available for figure 5:

**Figure supplement 1.** MRC5 cells were uninfected or infected with HSV-1 for 2 h, and ChIP-Seq for ICP4, Pol II, H3, H3K4me3, H3K27acetyl, H3K9me3, and H3K27me3 was performed.

DOI: https://doi.org/10.7554/eLife.51109.011

**Figure supplement 2.** MRC5 cells were infected with HSV-1 for 2 h, and ChIP-Seq for ICP4, Pol II, H3, H3K4me3, H3K27acetyl, H3K9me3, and H3K27me3 was performed.

DOI: https://doi.org/10.7554/eLife.51109.012

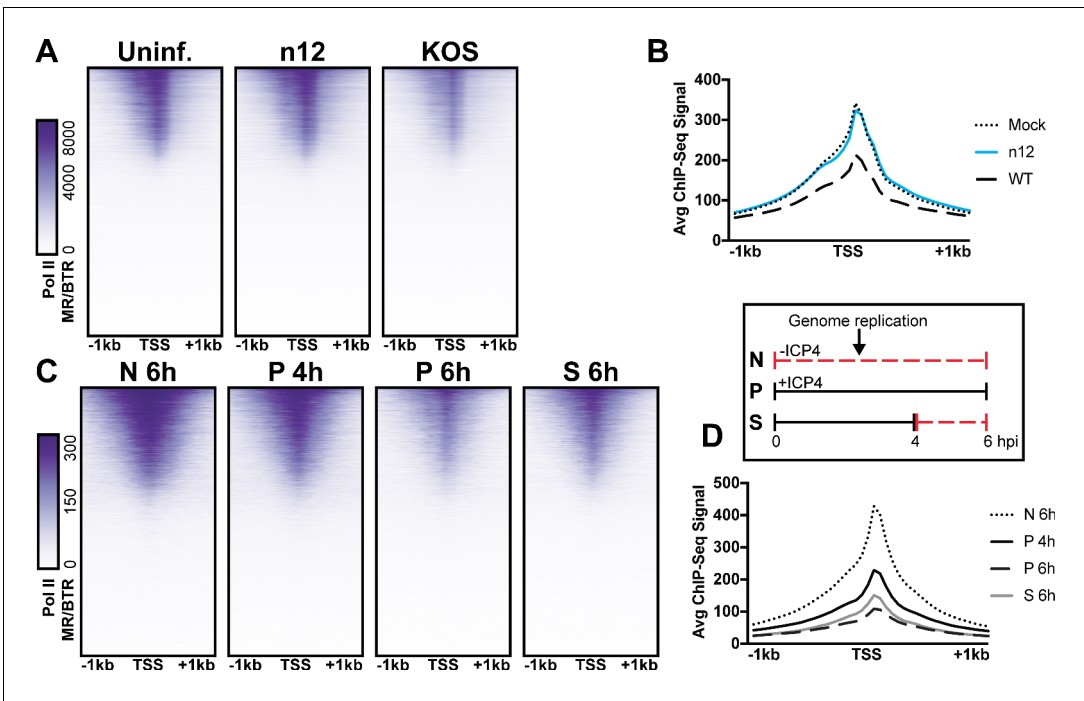

**Figure 6.** The role of ICP4 in Pol II loss on host promoters. MRC5 cells were (**A-B**) uninfected or infected with n12 or WT HSV-1 and harvested prior to genome replication or C-D) infected with tsKos and grown at permissive conditions (P), shifted up from permissive to nonopermissive conditions at four hpi (S), or nonpermissive conditions (N). (**A–D**) ChIP-Seq for Pol II was performed. Data was aligned to the human genome (hg38) and normalized for cellular genome sampling. ChIP-Seq data was aligned to cellular promoters + /- 1 kilobase from TSS. The average signal for each condition plotted as a line graph.

DOI: https://doi.org/10.7554/eLife.51109.013

to mock-infection, as we just established that Pol II depletion does not occur in n12 infection (*Figure 6A–B*). We observed significant depletion of Pol II from cellular promoters in permissive and shifted samples (*Figure 6C–D*). Pol II depletion was not directly related to viral genome copy number. tsKos shifted up had the highest number of viral genomes present, but did not reach the same level of cellular Pol II depletion as cells grown at permissive temperature for the same length of time. These data suggest that the viral genome is not solely responsible for preferential recruitment of cellular Pol II. Instead ICP4 bound to the viral genome is required for depletion of Pol II from cellular promoters. These results suggest a model in which genome replication facilitates host Pol II depletion when the relative number of ICP4 to viral genomes is high (2 hpi). As the number of ICP4 bound viral genomes increased, we observed a corresponding decrease in Pol II on host promoters.

## Discussion

### ICP4 as a sink for general transcription factors

ICP4 is synthesized shortly after the viral genome enters the nucleus and remains associated with the genome through all phases of infection. Our data demonstrated that ICP4 bound promiscuously to the viral genome prior to DNA replication. At this time point, the relative concentration of ICP4 to viral genomes was relatively high which likely promoted multimerization on DNA through ICP4-ICP4 interactions (*Kuddus and DeLuca, 2007*). We observed a similar phenotype for ICP4's interaction partner, Mediator (*Lester and DeLuca, 2011*; *Wagner and DeLuca, 2013*). Components of Mediator bound generally to the viral genome, concentrating near viral TATA boxes. Additional protein-Mediator interactions likely contribute to this distribution. This is a unique recruitment phenotype for Mediator which binds exclusively at cellular TSS via multiple protein-protein interactions. In the absence of ICP4, these interactions were not sufficient to support Mediator binding to the viral

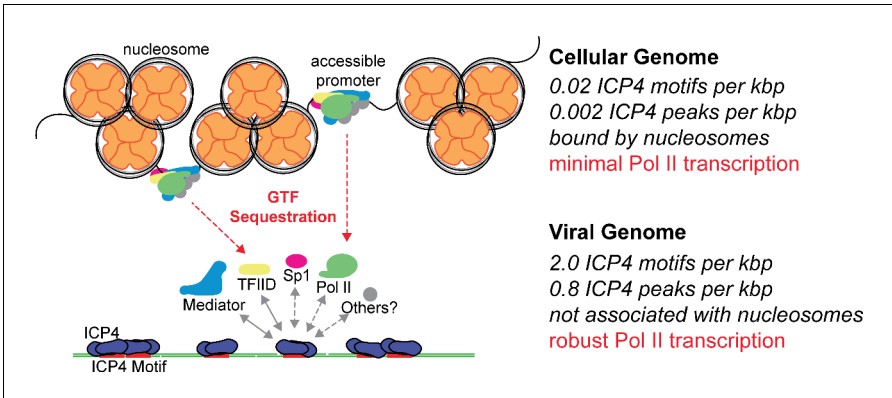

**Figure 7.** Model for ICP4 function. ICP4 preferentially binds to the more accessible viral genome recruiting cellular transcription factors preferentially to the viral genome, thus activating the virus and inhibiting cellular transcription. DOI: https://doi.org/10.7554/eLife.51109.014

genome. With the exception of Mediator recruitment to IE promoters which does not require ICP4 and reflects the activity of VP16 (*Batterson and Roizman, 1983*; *Campbell et al., 1984*). Similarly, we observed a 2 to 10-fold decrease in recruitment of Pol II, TBP, and Sp1 to viral E and L promoters without ICP4. This minimal level of recruitment is insufficient to support transcription, which explains why only IE transcripts are efficiently transcribed in the absence of ICP4.

ICP4-dependent GTF recruitment was not due to a global accessibility change. In the absence of ICP4 the viral genome remained absent of histones and had little change in tagmentation frequency. This is most likely due to the action of ICP0, which is an IE protein expressed in the absence of ICP4 and has been shown to preclude histones from the genome (*Cliffe and Knipe, 2008*; *Ferenczy and DeLuca, 2009*). We posit that ICP4's ability to interact and recruit Mediator and TFIID generally to the viral genome creates a local concentration gradient. Ultimately this increases the incidence of Pol II transcription machinery recruitment to the viral genome, which is stabilized by contact with additional protein-DNA, protein-protein interactions. Our observation that ICP4 coats the viral genome, a unique recruitment phenotype for a protein that functions in GTF recruitment likely reflects the architecture of the viral genome. The condensed organization of the HSV genome results in overlap of viral promoters and ORFS, with little to no unused coding space. For these reasons, it may be advantageous for ICP4 to generally and dynamically bind to the viral genome. It is also possible that the relatively high density of ICP4 on the viral genome prior to DNA replication may serve to repress transcription of the true late promoters, which only contain binding sites for the core GTFs. It has been shown that ICP4 binding can impose an increased dependence on DNA replication for expression from relatively simple promoters (*Koop et al., 1993*; *Rivera-Gonzalez et al., 1994*). These data demonstrated the critical role ICP4 serves as a general viral transcription factor, essential for activation and continued transcription of E and L genes.

## ICP4 differentiates between the viral and cellular genome

ICP4 possesses the ability to bind to double stranded DNA independent of sequence, an ability that is facilitated by ICP4 oligomerization on the genome. At early time points, when the relative concentration of ICP4 to the viral genome was high, we observed promiscuous ICP4 binding. This coating phenotype provides an explanation for why no specific binding sites on the genome affect the ability of ICP4 to activate transcription (*Coen et al., 1986*; *Smiley et al., 1992*). Instead the high density of ICP4 binding motifs on the viral genome aggregate to create a global affinity for ICP4.

Early during infection, we also observed binding of ICP4 to cellular promoters, a novel observation. ICP4 bound promoters of genes which grouped ontologically to common housekeeping functions including pathways related to chromatin, transcription, and metabolism. Furthermore, we found that ICP4 specifically bound where there was an absence of histones, adjacent to euchromatic markers. ICP4 did not alter the chromatic markers of the promoters it bound to, rather ICP4 bound regions which were also accessible in the absence of infection. This likely reflects ICP4's ability to

bind naked double-stranded DNA in a sequence independent manner. This ability is promoted by ICP4 multimerization (*Kuddus and DeLuca, 2007*), which is likely why we see the binding early during infection. The GTF's present on these cellular promoters, namely TFIID and Mediator complex, may further stabilize ICP4 binding at these promoters via their well-characterized interactions (*Carrozza and DeLuca, 1996*; *Lester and DeLuca, 2011*; *Wagner and DeLuca, 2013*).

The consequences, if any, of this binding for the transcription of specific cellular genes remains to be determined. The binding of ICP4 to the cellular genome was greatly diminished by four hpi, which corresponded to 3–4 viral genome duplications. At this time point we also observed a decrease in ICP4 coating the viral genome. However, ICP4 still bound abundantly, concentrating adjacent to strong cognate binding sites. This is most likely due to replication of the viral genome producing more ICP4 binding targets. Simple mass action results in binding to predominantly higher affinity sites. We propose that the binding preference of ICP4 for the viral genome is due to the 100-fold higher density of cognate binding sites and absence of cellular histones.

## ICP4 as both viral transcription factor and chromatin

HSV-1 productive infection generates 1,000–10,000 viral progeny per infected cell within a 24 h window. To facilitate this rampant transcriptional shift HSV-1 manipulates host Pol II machinery to prioritize viral mRNAs. By six hpi viral mRNA's comprise almost 50% of the total mRNA present in the host nucleus (*Dremel and DeLuca, 2019*). Furthermore, binding of Pol II to cellular promoters dramatically decreases upon HSV-1 infection (*Abrisch et al., 2015*; *Birkenheuer et al., 2018*). A recent study concluded that viral replication compartments efficiently enrich Pol II into membraneless domains (*McSwiggen et al., 2019*). Herein we identified the viral factor responsible for coopting the host Pol II machinery.

McSwiggen et al. proposed this phenomenon was dependent on the absence of nucleosomes which made the viral genome 100-fold more accessible than the cellular genome. While we agree that this accessibility is critical for viral infection, we believe it is essential for ICP4 binding. Similar to cellular chromatin, ICP4 coats the viral genome throughout productive infection (*Figure 7*). However, ICP4 also functions to scaffold Pol II transcription machinery to the viral genome. We demonstrated that Pol II depletion from cellular promoters was dependent on the number of ICP4 bound viral genomes. We propose that one or more components of the PIC, such as the ICP4 binding partners TFIID and Mediator, are limiting and ICP4 recruits these factors to the viral genome. As the number of viral genomes bound by ICP4 increases, the limiting PIC components no longer contacts cellular promoters. Ultimately this results in decreased Pol II occupancy on host promoters, preventing cellular transcription. This mechanism is essential to facilitate the rapidly progressing infection while limiting the extent to which the host can respond to viral challenge.

# Materials and methods

**Key resources table**

| Reagent type (species) or resource | Designation | Source or reference | Identifiers | Additional information |
|---|---|---|---|---|
| Cell line (*Homo-sapiens*) | MRC5 | ATCC | ATCC CCL-171 RRID:CVCL_0440 | |
| Cell line (*Cercopithecus aethiops*) | Vero | ATCC | ATCC CCL-81 RRID:CVCL_0059 | |
| Cell line (*Cercopithecus aethiops*) | E5 | *DeLuca and Schaffer, 1988* | E5 | Vero cells stably expressing ICP4 |
| Strain, strain background (*Human herpesvirus 1*) | KOS | *Smith, 1964* | | |

*Continued on next page*

*Continued*

| Reagent type (species) or resource | Designation | Source or reference | Identifiers | Additional information |
|---|---|---|---|---|
| Strain, strain background (*Human herpesvirus 1*) | n12 | *DeLuca and Schaffer, 1988* | Strain KOS with nonsense mutation near ICP4 N-terminus | |
| Strain, strain background (*Human herpesvirus 1*) | tsKos | *Dremel and DeLuca, 2019* | Strain KOS with temperature sensitive ICP4 | ICP4 has an A474V mutation |
| Antibody | Anti-RNA polymerase II CTD repeat YSPTSPS (phospho S5) Mouse Monoclonal | Abcam | Cat #ab5408 | |
| Antibody | Anti-TATA binding protein Mouse Monoclonal | Abcam | Cat #ab51841 | |
| Antibody | Anti-Sp1 Mouse Monoclonal | Santa Cruz | Cat #sc-17824 | |
| Antibody | Anti-Human Sur-2 (Med23) Mouse Monoclonal | BD Pharmingen | Cat #550429 | |
| Antibody | Anti-Med1 Rabbit Polyclonal | Bethyl Laboratories | Cat #A300-793A | |
| Antibody | Anti-Histone H3 (tri methyl K4) Mouse Monoclonal | Abcam | Cat #ab12209 | |
| Antibody | Anti-Histone H3 (tri methyl K27) antibody Mouse Monoclonal | Abcam | Cat #ab6002 | |
| Antibody | Anti-Histone H3 (acetyl K27) Rabbit Polyclonal | Abcam | Cat #ab4729 | |
| Antibody | Recombinant Anti-Histone H3 (tri methyl K9) Rabbit Monoclonal | Abcam | Cat #ab176916 | |
| Antibody | Anti-Histone H3 antibody Rabbit Polyclonal | Abcam | Cat #ab1791 | |
| Antibody | Anti-ICP4 (58S) Mouse Monoclonal | Neal DeLuca | Hybridomas available from ATCC HB-8183 | |

## Cells and viruses

Vero (African green monkey kidney) and MRC5 (human fetal lung) cells were obtained from and propagated as recommended by ATCC. Viruses used in this study include n12 (*DeLuca and Schaffer, 1988*), tsKos (*Dremel and DeLuca, 2019*) and KOS (*Smith, 1964*). n12 virus stocks were prepared and titered in a Vero-based ICP4 complementing cell line, E5. E5 cells were generated by stable transfection of Vero cells with ICP4 gene encoded on pK1-2 (GenBank Nucleotide JQ407535.1). E5 cells originate from a single colony, confirmed for its ability to complement the n12 virus and Western Blot assessed ICP4 expression (*DeLuca and Schaffer, 1988*). KOS virus stocks were prepared and titered in Vero cells. tsKos virus stocks were prepared and tittered in Vero cells at permissive temperature (33.5°C). All cells are regularly tested for the presence of mycoplasma contamination. Cells used in this study were mycoplasma free.

## Antibodies

The following antibodies were used: Pol II 4H8 specific to phospho S5 of the C-terminal domain repeat region (AbCam #ab5408), TBP (AbCam #ab51841), Sp1 (SantaCruz #sc-17824), Med23 (BD Pharmingen #550429), Med1 (Bethyl #A300-793A), H3K4me3 (Abcam #ab12209), H3K27me3 (AbCam #ab6002), H3K27acetyl (AbCam #ab4729), H3K9me3 (AbCam #ab176916), H3 (AbCam #1791), and ICP4 58S (derived from hybridomas-ATCC HB8183).

## Viral infection

MRC5 cells were infected with 10 PFU per cell. Virus was adsorbed in tricine-buffered saline (TBS) for 1 hr at room temperature. Viral inoculum was removed, and cells were washed quickly with TBS before adding 2% FBS media. Infected samples were incubated at 37°C unless otherwise specified.

## ChIP-Sequencing

Infected cells were treated with 5 mL of 25% formaldehyde for 15 min at room temperature, followed by 5 mL of 2.5 M glycine. All following steps were performed at 4°C unless otherwise stated. Cultures were washed with TBS and scraped into 50 mL of FLB [5 mM 1,4-Piperazinediethanesulfonic acid (PIPES) pH 8, 85 mM KCl, 0.5% Igepal CA-630, 1x Roche protease inhibitor cocktail]. Cells were pelleted by low-speed centrifugation, resuspended in 1.1 mL RIPA buffer [1x phosphate-buffered saline (PBS), 0.5% sodium deoxycholate, 0.1% sodium dodecyl sulfate (SDS), 1x Roche protease inhibitor cocktail]. Sample was sonicated for 6 intervals of 30 s with a Sonics Vibra-Cell VCX 130 sonicator equipped with a 3 mm microprobe and pelleted at 2000 xg for 15 min. 50 µl was stored as an input control, and the remainder was divided equally to use in immunoprecipitations (IP). $2-4 \times 10^7$ MRC5 cells were applied per IP. Samples were immunoprecipitated with 25 µg (TBP, Sp1), or 10 µg (Pol II, H3K4me3, H3K27me3, H3K27ac, H3K9me3, H3) antibody. Antibody was previously bound to 50 µL of Dynabeads M280 sheep anti-mouse IgG beads, or Dynabeads M280 sheep anti-rabbit IgG beads in 5% bovine serum albumin (BSA) 1x PBS overnight. DNA samples were bound to the antibody-bead complex overnight rotating. The IP mixtures were washed seven times with LiCl wash buffer [100 mM Tris-HCl buffer pH 7.5, 500 mM LiCl, 1% Igepal CA-630, 1% sodium deoxycholate] and once with Tris-EDTA (TE) buffer. Beads were resuspended in IP elution buffer [1% SDS, 0.1M NaHCO$_3$] and incubated at 65°C for 2 hr 900 rpm. Input aliquot was suspended in IP elution buffer. Input and IP samples were incubated at 65°C 900 rpm overnight. The samples were extracted with phenol-chloroform-isoamyl alcohol (25:24:1) and with chloroform-isoamyl alcohol (24:1) and then purified using Qiagen PCR cleanup columns. Each sample was quantified using a Qubit 2.0 fluorometer (Invitrogen) and 2–20 ng was used to create sequencing libraries using the NEBNext Ultra II DNA Library preparation kit (NEB #E7103S). Libraries were quantified using the Agilent DNA 7500 Kit, and samples were mixed together at equimolar concentration. Illumina HiSeq 2500 single-end 50 bp sequencing was carried out at the Tufts University Core Facility.

## ATAC-Sequencing

We adapted the protocol from *Buenrostro et al. (2013)*. Briefly, 2 million MRC5 cells were plated into 60 mm dishes and allowed to grow overnight. Cells were infected as described above. Uninfected and n12 infected cells were harvested at four hpi. WT HSV-1 infected cells were harvested pre-replication at two hpi. Infected samples were washed once with chilled TBS and lysis-1 buffer [10 mM Tris-HCl pH 7.4, 10 mM NaCl, 3 mM MgCl$_2$]. Samples were incubated with 2 mL lysis-2 buffer [10 mM Tris-HCl pH 7.4, 10 mM NaCl, 3 mM MgCl$_2$, 0.1% Igepal CA-630] for 3 min on ice. Cells were gently resuspended and dounced until nuclei were visible via trypan blue staining. Nuclei were spun at 500 g for 10 min at 4°C and resuspended in lysis-1 buffer. 250 µL ($5 \times 10^5$ cells) was transferred to an epindorf tube and spun at 500 g for 10 min at 4°C. Nuclei were resuspended in 22 µL buffer TD (Illumina Catalog No. 15027866) 2.5 µL TDE1 (Illumina Catalog No. 15027865) and 22.5 µL water and incubated at 37°C for 30 min gently shaking. DNA was purified using the MinElute PCR purification kit (Qiagen Cat No./ID: 28004). PCR amplification was performed for 8–14 total cycles. Libraries were quantified using the Agilent DNA 7500 Kit, and samples were mixed together at equimolar concentration. Sequencing was carried out at the Tufts University Core Facility, Illumina HiSeq 2500 single-end 50 bp sequencing was carried out for replicate 1, Illumina Mid-Output NextSeq was carried out for replicates 2 and 3.

## Data analysis

### ChIP-Seq

Data was uploaded to the Galaxy web platform, and we used the public server at usegalaxy.org to analyze the data (**Afgan et al., 2018**). Data was first aligned using Bowtie2 (**Langmead and Salzberg, 2012**) to the human genome (hg38), and then unaligned reads were mapped to the HSV-1 strain KOS genome (KT899744.1). Bam files were visualized using DeepTools bamcoverage (**Ramírez et al., 2016**) with a bin size of 1 to generate bigwig files. Data was viewed in IGV viewer and exported as EPS files. Bigwig files were normalized for sequencing depth and genome quantity. Mapped reads were multiplied by the 'norm factor' which was calculated as the inverse of $(Input\ cellular\ reads)/(Input\ cellular\ +\ viral\ mapped\ reads\ (TMR)) \times\ Billion\ sample\ TMR$ or $(Input\ viral\ reads\ )/TMR \times\ Million\ sample\ TMR$. ChIP-Seq experiments were repeated for a total of 2 to 4 biological replicates. The normalized bigwig files were averaged between replicates. Heatmaps and gene profiles were generated using MultiBigwigSummary (**Ramírez et al., 2016**) on normalized cellular bigwig files to all UCSC annotated mRNAs. Gene profiles and heatmaps were plotted using plotProfile and plotHeatmap (**Ramírez et al., 2016**). Spearman correlation analysis was performed using deeptools plotCorrelation on multiBigwigSummary limited to cellular transcripts (**Ramírez et al., 2016**).

### Peak calling

Viral peaks were called using MACS2 call peak (**Feng et al., 2012**), pooling treatment and control files for each condition. Due to the small size of the viral genome (151974 bp) we could not use the shifting model option (–nomodel). To offset the dense binding of ICP4 we used a fixed background lambda as local lambda for every peak region and a more sophisticated signal processing approach to find subpeak summits in each enriched peak region (–call-summits).

Cellular peaks were called using MACS2 (**Feng et al., 2012**). We first removed non-uniquely mapped sequences with SAMtools, filter SAM or BAM for a minimum MAPQ quality score of 20 (**Li et al., 2009**). We determined the approximate extension size for each IP using MACS2 predictd, and averaging the size estimate between replicates. We ran MACS2 call peak for individual replicates and pooled samples with no shifting model (–nomodel). To determine high confidence peaks present in each MACS2 output we used Galaxy Operate on Genomic Intervals, Join. Peak intersection was analyzed for intersection size and jaccard statistic using JaccardBed (**Ramírez et al., 2016**). ChIPseeker was run on MACS2 outputs to assess the cellular regions bound in each condition (**Yu et al., 2015**).

### Motif discovery

Bedtools Multiple Intersect (**Quinlan and Hall, 2010**) was used to compare the MACS2 output for ICP4 IP at 2, 4, and 6 hr. A BED file was generated for regions + /- 100 bp from the summits of each identified peak. Peaks in common between all three experimental conditions were used to generate a fasta file using GetFastaBed (**Quinlan and Hall, 2010**) Peaks present in all three time points were submitted to MEME v.4.11.1.0 for motif analysis (**Bailey et al., 2009**). The consensus sequence in **Figure 1** had the most significant E-value, and was the only motif found in more than five peaks.

### Correlation analysis

To assess quality and reproducibility of data we assessed normalized bigwig files for each IP replicate. For cellular and viral alignments we ran MultiBigwigSummary (**Ramírez et al., 2016**) with a bin size of 10,000 and 50 bp, respectively. Raw bin counts were plotted and a linear regression analysis was performed (**Figure 3—figure supplements 1–2**).

### ATAC-Seq

Data was first aligned using Bowtie2 (**Langmead and Salzberg, 2012**) to the human genome (hg38), and then unaligned reads were mapped to the HSV-1 strain KOS genome (KT899744.1) with the following parameters: –no-unal –local –very-sensitive-local –nodiscordant –no-mixed –contain –overlap –dovetail –phred33. Approximate fragment size for paired end read data was determined from mapped bam files using CollectInsertSizeMetrics (Broad Institute, Picard). Bam files were visualized using DeepTools bamcoverage (**Ramírez et al., 2016**) with a bin size of 1 to generate bigwig files.

Data was viewed in IGV viewer and exported as EPS files. Cellular bigwig files were normalized for sequencing depth, the y-axes values are mapped reads per billion total reads. Viral bigwig files were further normalized per viral genome copy number, see Table 3C. The y-axes values are mapped reads per billion total reads per viral genome copy number. The normalized bigwig files were averaged between three biological replicates. Heatmaps and gene profiles were generated using Multi-BigwigSummary (*Ramírez et al., 2016*) on normalized cellular bigwig files to all UCSC annotated mRNAs. Gene profiles and heatmaps were plotted using plotProfile and plotHeatmap (*Ramírez et al., 2016*). To calculate the percentage of total DNA corresponding to the virus or host in n12 and WT HSV-1 infection, we utilized ChIP-Seq input reads. We calculated the average percentage of total reads which mapped to either the virus or host in four biological replication ChIP-Seq samples. We used this value to calculate the number of viral genomes contained within each nucleus. This value was used to determine the tagmentation enrichment observed relative to the actual amount of genome content present.

## Acknowledgements

This work was supported by NIH grants AI030612 and AI143179 to NAD. SED was supported by the NIH training grants T32AI060525 and F31AI36251. We acknowledge members of the DeLuca lab for thoughtful discussions related to this project and Frances Sivrich for technical assistance.

## Additional information

### Funding

| Funder | Grant reference number | Author |
|---|---|---|
| National Institute of Allergy and Infectious Diseases | R01AI30612 | Neal A DeLuca |
| National Institute of Allergy and Infectious Diseases | F31AI36251 | Sarah E Dremel |
| National Institute of Allergy and Infectious Diseases | T32AI060525 | Sarah E Dremel |
| National Institute of Allergy and Infectious Diseases | R21AI143179 | Neal A DeLuca |

The funders had no role in study design, data collection and interpretation, or the decision to submit the work for publication.

### Author contributions

Sarah E Dremel, Conceptualization, Data curation, Formal analysis, Funding acquisition, Investigation, Methodology, Writing—original draft, Writing—review and editing; Neal A DeLuca, Conceptualization, Supervision, Funding acquisition, Investigation, Methodology, Writing—original draft, Writing—review and editing

### Author ORCIDs

Sarah E Dremel https://orcid.org/0000-0003-0968-3090
Neal A DeLuca https://orcid.org/0000-0001-8381-8577

### Decision letter and Author response

Decision letter https://doi.org/10.7554/eLife.51109.027
Author response https://doi.org/10.7554/eLife.51109.028

## Additional files

### Supplementary files

• Transparent reporting form DOI: https://doi.org/10.7554/eLife.51109.015

## Data availability

All data are publicly accessible in the SRA database (PRJNA553543, PRJNA553555, PRJNA553559, PRJNA553563, PRJNA508791).

The following datasets were generated:

| Author(s) | Year | Dataset title | Dataset URL | Database and Identifier |
|---|---|---|---|---|
| Sarah E Dremel, Neal A DeLuca | 2019 | ICP4 ChIP-Seq of WT (KOS) HSV-1 Productive Infection in MRC5 cells | https://www.ncbi.nlm.nih.gov/bioproject/PRJNA553563 | Sequence Read Archive, PRJNA553563 |
| Sarah Dremel, Neal DeLuca | 2019 | ATAC-Seq of WT (KOS) and ICP4 null (n12) HSV-1 Productive Infection in MRC5 cells | https://www.ncbi.nlm.nih.gov/bioproject/PRJNA553559 | Sequence Read Archive, PRJNA553559 |
| Sarah Dremel, Neal DeLuca | 2019 | GTF ChIP-Seq of WT (KOS) and ICP4 null (n12) HSV-1 Productive Infection in MRC5 cells | https://www.ncbi.nlm.nih.gov/bioproject/PRJNA553555 | Sequence Read Archive, PRJNA553555 |
| Sarah E Dremel, Neal A DeLuca | 2019 | Histone ChIP-Seq of WT (KOS) and ICP4 null (n12) HSV-1 Productive Infection in MRC5 cells | https://www.ncbi.nlm.nih.gov/bioproject/PRJNA553543 | Sequence Read Archive, PRJNA553543 |

The following previously published dataset was used:

| Author(s) | Year | Dataset title | Dataset URL | Database and Identifier |
|---|---|---|---|---|
| Sarah E Dremel, Neal A DeLuca | 2019 | ChIP-Seq of HSV-1 (tsKos) Productive Infection | https://www.ncbi.nlm.nih.gov/bioproject/PRJNA508791 | Sequence Read Archive, PRJNA508791 |

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
