## [Decision Letter]

Thank you for submitting your article "Herpes Simplex nucleoprotein creates a competitive environment facilitating robust viral transcription and host shut off" for consideration by *eLife*. Your article has been reviewed by three peer reviewers, one of whom is a member of our Board of Reviewing Editors, and the evaluation has been overseen by Jessica Tyler as the Senior Editor. The following individual involved in review of your submission has agreed to reveal their identity: Lars Dölken (Reviewer #2).

The reviewers have discussed the reviews with one another and the Reviewing Editor has drafted this decision to help you prepare a revised submission.

Summary:

The manuscript describes how the HSV-1 protein ICP4 plays a pivotal role in the reprogramming of the cellular Pol II transcription machinery. The authors show that the ICP4 preferentially binds to the viral genome and forms a viral nucleoprotein, which then preferentially recruits Pol II transcription factors to the viral genome, effectively causing the host-shut off. Apparently, ICP4 favorably binds to the incoming naked viral DNA at early times of infection. Overall, the paper describes a novel and important property of ICP4 protein in the life cycle of HSV. However, the reviewers raised a number of questions that should be addressed in the revised submission.

Essential revisions:

1) In ICP4 null mutants, one can still observe relatively high read coverage for ICP4 binding (e. g. Figure 2—figure supplement 1, and others). How do the authors explain this pattern for ICP4 null mutant, where no ICP4 binding should be observed? If such a coverage represents non-specific background, the authors should explicitly state whether the background reads have been subtracted from the coverage graphs of ICP4 binding in all figures?

2) The main conclusion of the authors is that ICP4 not only binds a specific motif at viral E promoters early in infection but promiscuously binds the incoming viral DNA. While the quantitative analysis strongly support their hypothesis, other explanations and particularly the implications of their finding should be considered and more thoroughly discussed. Of note, the effect of ICP4 on viral genome accessibility was rather small (2.8- vs. 4-fold). What is the advantage for the virus recruiting ICP4 in a sequence-independent manner to the viral genome? How does this contribute to differences in E and L promoters? The authors should expand their discussion on this.

3) The authors state that ICP4 binding to the cellular genome is restricted to accessible regions of the cellular genome, namely the promoter regions. However, ICP4 is well known to bind to Mediator, which shows the same distribution pattern as ICP4 on cellular genes early in infection as ICP4. As specific ICP4 binding motifs do not explain the binding of ICP4 to cellular promoter regions, an alternative explanation is that ICP4 is recruited to cellular promoter regions via its interaction with Mediator (and not by its DNA-binding activity). ATAC-seq also reveals sites of the genome which become "accessible" independent of Mediator. The authors should check for co-localisation of ICP4-reads with ATAC-seq peaks outside of promoter regions (with no overlapping Mediator reads).

4) The antibodies section of the Materials and methods should indicate that the Pol II antibody is specific for phosphorylated Serine 5 residues of the CTD. The use of the 4H8 clone of the Pol II antibody as a readout for total pol II occupancy should generally be avoided. While this clone is commonly used to measure global pol II occupancy, phospho-Serine 5 is enriched compared to total pol II at promoters and splice sites. This bias, if anything, was likely beneficial to the authors as they focus the most on promoters, though it may preclude more in-depth analysis in the future in regards to other steps of transcription or under experimental conditions in which Serine 5 phosphorylation is altered. Thus, it is important for this bias (stated on the manufacturer's website) to be apparent at-a-glance to the reader.

5) The authors adapted the y-axis scale in a number of figures to provide a better visualization of their data. While this was done well for most figures, some of the scale adjustments make it hard to compare results. For example in Figure 2B, the authors show the occupancy of polII, TBP aso. on two selected virus genes. Unfortunately, the y-axis drastically vary between the panel, making a comparison very difficult (e.g. Sp1 32,312 vs. 11,877). These differences make it also hard to compare the n12 mutant and wild type virus in Figure 3—figure supplement 3. The authors should standardize the y-axis scales to allow the comparison between panels and to avoid miss-interpretations.

6) Subsection “ICP4 as both viral transcription factor and chromatin”: It is currently unclear whether the recruitment of major IE transcription factors to the incoming viral genomes only occurs in HSV infection or also in other herpesviruses, e.g. CMV infection. Therefore, there is currently no evidence that this explains differences in the kinetics of lytic replication between different herpesviruses. Interestingly, the relative contribution of viral gene expression during the first 2h of MCMV infection is even higher than observed for HSV-1. However, MCMV DNA replication does not start until 12 h p.i. whereas it initiates already at 2 h p.i. in HSV-1 infection. Two hours of interferon treatment are not sufficient to induce a strong protective antiviral state. Therefore, host responses are unlikely to prevent viral DNA replication in MCMV infection from already initiating at 2 h p.i. Combined, this strongly argues against a prominent role of the recruitment of ICP4 to viral genomes in explaining the differences in replication kinetics between different herpesviruses. Considering the conserved IE, E, L gene expression cascade, one might assume that coating of the incoming viral genomes with the major viral transactivator is a conserved feature of herpesviruses.

---

## [Author Response]

Essential revisions:1) In ICP4 null mutants, one can still observe relatively high read coverage for ICP4 binding (e. g. Figure 2—figure supplement 1, and others). How do the authors explain this pattern for ICP4 null mutant, where no ICP4 binding should be observed?

ICP4 is packaged into the viral tegument and thus enters the cell with the infecting virion. In order to produce stocks of ICP4 null virus, we must grow the virus in complementing cells which produce ICP4. Thus, n12 still has ICP4 packaged into the virions, but does not produce any nascent ICP4 during infection. We cannot say whether the reads we see with ICP4 ChIP-seq of n12 infection are due to incoming tegument ICP4 binding the viral genome, or ICP4 bringing viral genome fragments into the tegument during virion production in the complementing cells. To clarify this fact for the readers we have added the following to the text:

“There were a relatively small number of reads in the ICP4 ChIP of n12 (Figure 2A, Figure 2—figure supplement 1). […] While the ultimate source of these reads is not clear at present, the amount of binding of ICP4 from the virion to DNA is not sufficient to promote transcription complexes on viral early and late genes.”

If such a coverage represents non-specific background, the authors should explicitly state whether the background reads have been subtracted from the coverage graphs of ICP4 binding in all figures?

We did not subtract input or background reads from any of our ChIP-seq data. The data has been multiplied by a set factor in order to adjust for sequencing depth and viral genome copy number (see additional details in the Materials and methods section).

2) The main conclusion of the authors is that ICP4 not only binds a specific motif at viral E promoters early in infection but promiscuously binds the incoming viral DNA. While the quantitative analysis strongly support their hypothesis, other explanations and particularly the implications of their finding should be considered and more thoroughly discussed. Of note, the effect of ICP4 on viral genome accessibility was rather small (2.8- vs. 4-fold).

We have added a panel to Figure 3 to demonstrate that there is no difference in nucleosome occupancy in the absence of ICP4. Additionally, we repeated the ATAC-Seq experiment with two additional biological replicates, which confirmed the original data. We have now adjusted Figure 3 to include all the biological replicates. The calculations in Table 3C has been adjusted to include the new data. We now observe that n12 and WT virus have a tagmentation enrichment of 2.2 and 2-fold, respectively.

What is the advantage for the virus recruiting ICP4 in a sequence-independent manner to the viral genome? How does this contribute to differences in E and L promoters? The authors should expand their discussion on this.

The following has been added:

“Our observation that ICP4 coats the viral genome, a unique recruitment phenotype for a protein that functions in GTF recruitment likely reflects the architecture of the viral genome. […] It is also possible that the relatively high density of ICP4 on the viral genome prior to DNA replication may serve to repress transcription of the true late promoters, which only contain binding sites for the core GTFs. It has been shown that ICP4 binding can impose an increased dependence on DNA replication for expression from relatively simple promoters (Koop et al., 1993; Rivera-Gonzalez et al., 1994).”

3) The authors state that ICP4 binding to the cellular genome is restricted to accessible regions of the cellular genome, namely the promoter regions. However, ICP4 is well known to bind to Mediator, which shows the same distribution pattern as ICP4 on cellular genes early in infection as ICP4. As specific ICP4 binding motifs do not explain the binding of ICP4 to cellular promoter regions, an alternative explanation is that ICP4 is recruited to cellular promoter regions via its interaction with Mediator (and not by its DNA-binding activity).

We agree that ICP4 may be stabilized in binding to cellular promoters via it’s well known interactions with Mediator and TFIID components. However, this does not detract from ICP4’s ability to directly bind dsDNA which has been well established in vivo and in vitro. We have added the following:

“Furthermore, we found that ICP4 specifically bound where there was an absence of histones, adjacent to euchromatic markers. […] The GTF’s present on these cellular promoters, namely TFIID and Mediator complex, may further stabilize ICP4 binding at these promoters via their well-characterized interactions (Carrozza and DeLuca, 1996; Lester and Deluca; Wagner and DeLuca, 2013).”

ATAC-seq also reveals sites of the genome which become "accessible" independent of Mediator. The authors should check for co-localisation of ICP4-reads with ATAC-seq peaks outside of promoter regions (with no overlapping Mediator reads).

We realized that Figure 5—figure supplement 2 was not included in the PDF sent to reviewers, we have corrected this. In Figure 5—figure supplement 2 we show that ICP4 peaks were primarily localized (>80%) within 1 kilobase of mRNA promoters, referenced in the subsection “ICP4 binding is restricted to accessible regions of the cellular genome”. For this reason, we limited our analyses within the paper to promoter regions.

4) The antibodies section of the Materials and methods should indicate that the Pol II antibody is specific for phosphorylated Serine 5 residues of the CTD. The use of the 4H8 clone of the Pol II antibody as a readout for total pol II occupancy should generally be avoided. While this clone is commonly used to measure global pol II occupancy, phospho-Serine 5 is enriched compared to total pol II at promoters and splice sites. This bias, if anything, was likely beneficial to the authors as they focus the most on promoters, though it may preclude more in-depth analysis in the future in regards to other steps of transcription or under experimental conditions in which Serine 5 phosphorylation is altered. Thus, it is important for this bias (stated on the manufacturer's website) to be apparent at-a-glance to the reader.

We have added the following:

“In all Pol II IP’s we used an antibody which preferentially binds to phospho S5 of the C-terminal domain repeat region. This post-translational modification is associated with Pol II found on mRNA promoters and splice sites.”

“specific to phospho S5 of the C-terminal domain repeat region”

5) The authors adapted the y-axis scale in a number of figures to provide a better visualization of their data. While this was done well for most figures, some of the scale adjustments make it hard to compare results. For example in Figure 2B, the authors show the occupancy of polII, TBP aso. on two selected virus genes. Unfortunately, the y-axis drastically vary between the panel, making a comparison very difficult (e.g. Sp1 32,312 vs. 11,877). These differences make it also hard to compare the n12 mutant and wild type virus in Figure 3—figure supplement 3. The authors should standardize the y-axis scales to allow the comparison between panels and to avoid miss-interpretations.

Figure 2B, and Figure 3—figure supplement 3A have been adjusted so that the y-axes for each IP are the same across panels.

6) Subsection “ICP4 as both viral transcription factor and chromatin”: It is currently unclear whether the recruitment of major IE transcription factors to the incoming viral genomes only occurs in HSV infection or also in other herpesviruses, e.g. CMV infection. Therefore, there is currently no evidence that this explains differences in the kinetics of lytic replication between different herpesviruses. Interestingly, the relative contribution of viral gene expression during the first 2h of MCMV infection is even higher than observed for HSV-1. However, MCMV DNA replication does not start until 12 h p.i. whereas it initiates already at 2 h p.i. in HSV-1 infection. Two hours of interferon treatment are not sufficient to induce a strong protective antiviral state. Therefore, host responses are unlikely to prevent viral DNA replication in MCMV infection from already initiating at 2 h p.i. Combined, this strongly argues against a prominent role of the recruitment of ICP4 to viral genomes in explaining the differences in replication kinetics between different herpesviruses. Considering the conserved IE, E, L gene expression cascade, one might assume that coating of the incoming viral genomes with the major viral transactivator is a conserved feature of herpesviruses.

We have deleted the sentence pertaining to this at the end of the Discussion.